# Modified PID controller for automatic generation control of multi-source interconnected power system using fitness dependent optimizer algorithm

Amil Daraz[1]◉*, Suheel Abdullah Malik[1‡], Ihsan Ul Haq[1‡], Khan Bahadar Khan[2‡], Ghulam Fareed Laghari[1◉], Farhan Zafar[1◉]

**1** Department of Electrical Engineering, Faculty of Engineering and Technology, International Islamic University, Islamabad, Pakistan, **2** Department of Telecommunication Engineering, Faculty of Engineering and Technology, The Islamia University of Bahawalpur, Bahawalpur, Pakistan

◉ These authors contributed equally to this work.
‡ These authors also contributed equally to this work.
* amil.phdee108@iiu.edu.pk

**Data Availability Statement:** All relevant data are within the manuscript and its Supporting information files.

## Abstract

In this paper, a modified form of the Proportional Integral Derivative (PID) controller known as the Integral- Proportional Derivative (I-PD) controller is developed for Automatic Generation Control (AGC) of the two-area multi-source Interconnected Power System (IPS). Fitness Dependent Optimizer (FDO) algorithm is employed for the optimization of proposed controller with various performance criteria including Integral of Absolute Error (IAE), Integral of Time multiplied Absolute Error (ITAE), Integral of Time multiplied Square Error (ITSE), and Integral Square Error (ISE). The effectiveness of the proposed approach has been assessed on a two-area network with individual source including gas, hydro and reheat thermal unit and then collectively with all three sources. Further, to validate the efficacy of the proposed FDO based PID and I-PD controllers, comprehensive comparative performance is carried and compared with other controllers including Differential Evolution based PID (DE-PID) controller and Teaching Learning Based Optimization (TLBO) hybridized with Local Unimodal Sampling (LUS-PID) controller. The comparison of outcomes reveal that the proposed FDO based I-PD (FDO-I-PD) controller provides a significant improvement in respect of Overshoot (Osh), Settling time (Ts), and Undershoot (Ush). The robustness of an I-PD controller is also verified by varying parameter of the system and load variation.

## 1 Introduction

The electrical power system consist of highly complex structures having various network of varied loads are interconnected. The basic purpose of AGC is to provide desired amount of power within satisfactory quality to entire users. The system will be stable when there is an equilibrium between generated power and consumers load. Since, the consumers load

**Funding:** There is no specific grant for this Article.

**Competing interests:** The authors have declared that no competing interests exist.

normally changes, the active power drawn from the generator increases which reduce the speed of generator or turbine due to variation in frequency. The modern power system comprises of numerous network areas which are connected to transmission lines via tie-lines. AGC plays a key role to sustain the exchange of power between the control regions via tie-lines and retain the frequency at predetermined value [1, 2].

Automatic Generation Control (AGC) performs a significant contribution in maintaining the stability of the power system. In this regards, substantial consideration has been paid by researches to deal with the Load Frequency Control (LFC). However, initially, most of the work has been performed in the single area network. For instance, the authors in [3, 4] considered a single area network of hydro power generation and reheat thermal power respectively. On the other hand some of the authors worked in single source generation of multi- area IPS and utilized different control techniques. For example Satheeshkumar and Shivakumar in [5] used Ant Lion Optimization (ALO) method for the tuning of Proportional Integral (PI) controller of three area IPS. The authors consider single source of thermal generation units for area-1, hydro units for area-2 and Photo-voltaic (PV) for area-3. The outcomes yielded from the proposed method are compared with Genetic Algorithm (GA), Bat Inspired Algorithm (BIA) and PSO. Similarly, the works related to the single source multi-area are presented in reference [6–10].

In past few decades, numerous control techniques have been employed for the AGC of single as well as multi-area with multi-source IPS. Mohanty et al. [11] have deliberate LFC of single area with multi-source including hydro, gas and thermal units by employing I/PI/PID controllers. The DE optimization algorithm is considered in order to tune the proposed controller gains. The authors also extended the research work to two-area diverse power generation systems. The authors in [12] used TLBO with PD-PID controller for two- area network with multi-source system considering hydro, gas and thermal generation units. Patel et al. in [13] suggested a Fractional Order (FO) fuzzy PID controller for two area IPS considering three generation units in one area and three sources in other area. The gain of the controllers are optimized with Ant Lion Optimization (ALO) method.

The literature survey reveals that the nature-inspired meta-heuristic optimization algorithms have received tremendous attention from researchers due to their strengths and capabilities to solve numerous complex optimization problems in engineering. These techniques have also been successfully employed for the tuning of controller parameters. For instance, a Grouped Grey Wolf Optimizer (GGWO) [14] algorithm was used for the optimization of PI controller parameters on Doubly Fed Induction Generator (DFIG) based on wind turbines. In work [15], a Democratic Joint Operator Algorithm (DJOA) was applied for the tuning of PID controller gains of Permanent Magnetic Synchronous Generator (PMSG) considering wind energy conversion system. Authors in [16] considered photo-voltaic inverters based on solar energy harvesting by employing Ying Yang Pair Optimization (YYPO) algorithm to optimize the parameters of perturbation observer based fractional order PID controller. Similarly, Yang et al. [17] proposed a robust fractional order PID controller tuned with Interactive Teaching-Learning Optimizer (ITLO) for solving super capacitor energy storage system. Various researchers have attempt to solve the AGC problem by employing different meta-heuristic optimization algorithms. Of these methods, authors have utilized Genetic Algorithm (GA) [18], (PSO) [19], (DE) [20], Improved-Ant Colony Optimization (I-ACO) [21], Firefly Algorithms (FA) [22], Improved Grey Wolf Optimization (IGWO) [23], Teaching Learning Base Optimization (TLBO) [24], Symbiotic Organisms Search Algorithm (SOSA) [25], Salp Swarm Algorithm (SSA) [26], Imperialist Competitive Algorithm (ICA) [27], Sine Cosine Algorithm (SCA) [28] and Backtracking Search Algorithm (BSA) [29]. While some of the authors have also applied hybrid techniques like TLBO hybridized with Local Unimodal Sampling

(TLBO-LUS) [30], Improved Firefly with Pattern Search (IF-PS) [31], PSO hybrid with Gravitational Search Algorithm (PSO-GSA) [32], PSO hybridized with Chemical Reaction Optimization (HPSO-CRO) [33], and Hybrid PSO with Levy Flight algorithm (HPSO-LF) [34].

So far it remains a very challenging and critical task, due to the omnipresent problems of high dimensionality, non-differentiability and multimodality, to effectively and efficiently achieve the global optimum of engineering problems. Recently, a meta-heuristic algorithm called as Fitness Dependent Optimizer (FDO) has been developed to explore the swarming behavior of the bees. In short there are different types of bees: queen bees (responsible for making decision and produce offspring), worker bees (works under the command of queen bees) and scout bees (responsible for exploring environment and exploit the desirable targets) [35]. FDO algorithm has the advantages of fast convergence, simple to implement and adjust due to few parameters, higher efficiency and probability to find the global optimum and efficient exploration and exploitation. In this regards, FDO algorithm has been applied for the AGC problem to optimize the parameters of the proposed controller.

Initially, for the AGC problem integral controller has been widely used to control the load frequency of the system. But due to its slower response, the researchers have preferred to use PI controller which has the advantages of simple structure, low cost, fast response and easy to implementation [36]. The poor dynamic response of the PI controller has been improved by designing PID controller which has been widely used in industries due to its better performance, easy to design and stability [37, 38]. Similarly, the performance of PID controller has been improved by modifying the structure of PID controller without changing the system parameters. The modified form of PID controller known as I-PD controller has been successfully employed for the problem of magnetic levitation system [39], time delayed unstable process [40], and speed control of DC motor [41]. However, the literature reveals that modified form of PID controller has not been explored for the AGC problem. Therefore, in this paper the modified form of PID controller has been successfully applied.

In this paper, a novel modified form of PID controller known as I-PD controller is develop for AGC of multi-source IPS. In the proposed control model each area contains three different generation sources including hydro, reheat thermal and gas. Further, the parameters of the proposed controller is optimized with a more recent meta-heuristic algorithm known as Fitness Dependent Optimize (FDO). Moreover, the effectiveness of proposed approach has been assessed on two area network with four different scenarios i.e. reheat thermal, hydro, gas and multi-source power system. To quantify efficacy of the proposed controller's detailed comparative performance is made with the results obtained by DE-PID, TLBO-PID, LUS-PID, LUS-TLBO-PID, FA-PID, FA-I-PD, PSO-PID, PSO-I-PD and TLBO-I-PD. Further, various performance criteria including IAE, ITAE, ITSE, and ISE have been used to check the performance of the system. Finally, the robustness of I-PD controller has been verified by varying the system parameters within a range of ± 50%.

The respite of the research work is structured as follows: Section 2 represents the material and methods followed by controller structure and optimization technique, While in section 3 implementation and results are demonstrated. The presented research work is concluded with future direction in section 4.

## 2 Materials and methods

### 2.1 Controller structure

In early AGC method, the integral controller had been used to control the system frequency and tie-lines power. However, due to its slower time response, the researchers used Proportional Integral (PI) controller which have the advantages of its low cost, simple structure and

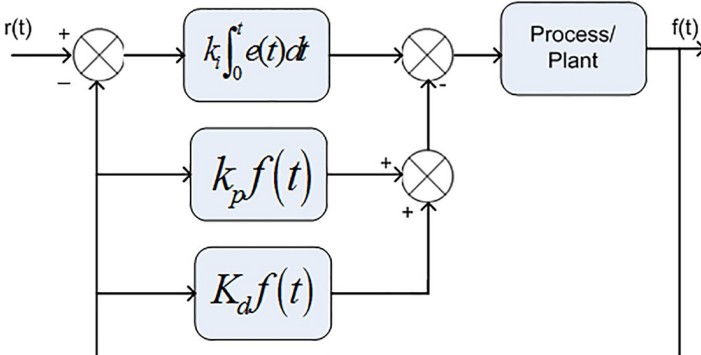

**Fig 1. Structure of I-PD controller.**

faster time response. The poor dynamic performance of PI controller has been improved by PID controller and their modified form I-PD controller which are nowadays very commonly used in practical [42, 43]. In I-PD controller, the proportional parameter and the derivative parameter are put in feedback form while, integrator parameter are put in feed forward direction, as depicted in Fig 1, while in PID controller all the parameters are put in feed forward direction which are depicted in Fig 2.

The input of PID /I-PD controller for area 1 and 2 is specified by Area Control Error (ACE).

$$ACE_1 \quad = \quad \beta_1 \Delta F_1 + \Delta P_{tie12} \tag{1a}$$

$$ACE_2 \quad = \quad \beta_1 \Delta F_2 + \Delta P_{tie21} \tag{1b}$$

where $\beta_1$ and $\beta_2$ represents biased parameters of frequency for area-1 and 2 respectively. $\Delta F_1$ and $\Delta F_2$ represents the frequency variation for area-1 and 2 respectively. Similarly, $\Delta P_{tie12}$ shows the tie line variation from area-1 to area- 2 and $\Delta P_{tie12}$ represent the tie line variation from area-2 to area-1.

To optimize the parameters of a controller, one of the essential steps is to determine the objective function. In this paper four different performance criteria, i.e., ITSE, ISE, ITAE and

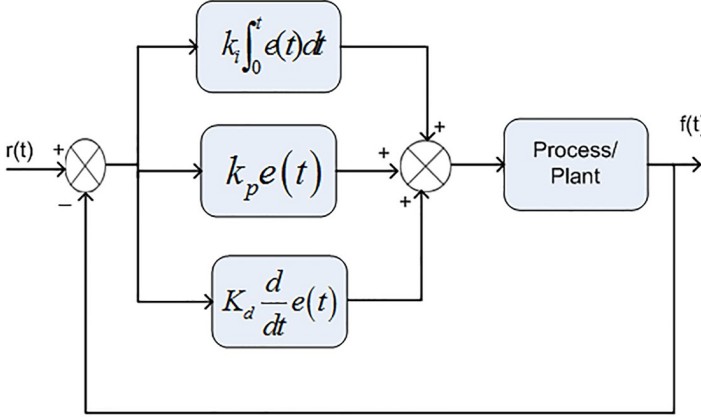

**Fig 2. Structure of PID controller.**

IAE are applied to verify the system performance and are given in below equations.

$$ITSE \quad = \quad \int_0^T t[\Delta F_1^2 + \Delta F_2^2 + \Delta P_{tie12}^2]dt \tag{2a}$$

$$ISE \quad = \quad \int_0^T [\Delta F_1^2 + \Delta F_2^2 + \Delta P_{tie12}^2]dt \tag{2b}$$

$$ITAE \quad = \quad \int_0^T t[|\Delta F_1| + |\Delta F_2| + |\Delta P_{tie12}|]dt \tag{2c}$$

$$IAE \quad = \quad \int_0^T [|\Delta F_1| + |\Delta F_2| + |\Delta P_{tie12}|]dt \tag{2d}$$

## 2.2 Fitness dependent optimizer

Several optimization methods are introduced by researchers in the field of power system to optimize the gains of the controllers by determination of fitness function. A nature-inspired meta-heuristic computational technique known as Fitness Dependent Optimizer (FDO) is deployed in this work to tune the various gains of the proposed controller. FDO starts from the initialization population of scout randomly in search space $K_m$; $m$ = 1, 2, 3, . . . . . ., $n$. The number of the scout bees were equal to population size and each scout contains three parameters known as $K_p$, $K_i$ and $K_d$ denote the gains of PID/I-PD Controller. Each scout bee denotes a new exposed hive (solution). Scouts bee are randomly searching more position to find best solutions. A previously discovered hive is ignored when new search space is find out. So each time the algorithm finds a better new solution the previous one is discarded. Moreover, if the scout bee does not find the best solutions by moving forward, then it comes back to its previous direction hoping for optimal solutions. However, if the prior solution is unable to provide a better new solution then the existing solution will be considered as the best solution that has been discovered yet. During random movement of the scout bees, each time adding pace to the present position, the scout hopes to determine the best solution. The results are compared with global best and hence it is repeated until the optimal solution not achieved or generation is stopped The movement of scouts is represented as below:

$$k_{m,t+1} = k_{m,t} + p, \tag{3}$$

where $m$ denotes current search agent, $t$ denotes current iteration, $K$ indicate scout bees and pace $p$ is the movement rate and direction. Generally, pace $p$ depends on Fitness Weight $w_f$ and can be articulated as follows:

$$w_f = |\frac{k_{m,t,f}^*}{k_{m,t,f}}| - \alpha \tag{4}$$

where $k_{m,t,f}^*$ represents the value of the best global solution, $k_{m,t,f}$ is the value of current solution and $\alpha$ is a weight factor which is used for the controlling of $w_f$ and its value is either 1 or 0. If the value of $\alpha$ is 1 then, it represent high level of convergence while if the value is 0 then there is no effect on Eq (4) but often it provides more stable search. The value of $w_f$ should be in the range of [1, 0]. But in some cases it may be equal to 1, for instance, when the current and global best solution are equal. The value of $w_f$ will be equal to 0 when $k_{m,t,f}^*$ is equal to 0. Finally, the

case $k_{m,t,f} = 0$ should be ignored. Hence, the rules given below must be considered:

$$p \quad = \quad Rk_{m,t,f}; \qquad \text{if} \qquad w_f = 0 \ OR \ w_f = 1 \ OR \ k_{m,t,f} = 0, \tag{5a}$$

$$p \quad = \quad \begin{cases} w_f(k_{m,t,f} - k^*_{m,t,f}) - 1 & \text{if} \qquad w_f < 1 \ AND \ w_f > 0 \ AND \ R < 0 \\ w_f(k_{m,t,f} - k^*_{m,t,f}) & \text{if} \qquad w_f < 1 \ AND \ w_f > 0 \ AND \ R \geq 0 \end{cases} \tag{5b}$$

where $R$ is the random number in the range of $[1, -1]$. The elementary steps of the FDO are shown in the algorithm 1.

　　**Algorithm 1**: Fitness Dependent Optimizer

```
Cost function J(.);
Generate scout bee population k_m,t; m = 1, 2, ···, n
While Boundary not reached do
for all For each scout bee k_m,t do
k*_m,t ← k_m,t
R ← r ∈ [-1 1] %random walk
if k_m,t,f == 0 then
w_f = 0
else
w_f ← |k*_m,t,f / k_m,t,f| − α
end if
if w_f == 0 OR w_f == 1 then
p ← R k_m,t
else
if R ≥ 0
p ←_f (k_m,t,f − k*_m,t,f) − 1
else
p ← w_f(k_m,t,f − k*_m,t,f)
end if
end if
K_m,t+1 ← K_m,t + p
If K_m,t+1,f < k_m,t,f then
Save k_m,t+1,f
Save p
else
K_m,t+1 ← K_m,t + p
if K_m,t+1,f < K_m,t,f then
Save k_m,t+1,f
Save p
else keep current position
endIf
endIf
end for
end while
```

## 3 Implementation and results

In this section FDO-I-PD and FDO-PID controllers are considered and employed for a two-area IPS with three different generation units to assess their performance. Further, to evaluate the performance of these controllers, a two-area IPS with a single generation unit including hydro, reheat thermal and gas is considered and then the same two-area network with all three sources is investigated via simulations with 1% step load perturbation (SLP). In prvious work,

**Table 1. Comparative performance for different indices criteria.**

| Controller with Techniques | Performance Indices | | | |
|---|---|---|---|---|
| | ITSE | ITAE | ISE | IAE |
| FDO-PID | 0.000056 | 0.000093 | 0.00067 | 0.0147 |
| FDO-I-PD | 0.000026 | 0.000019 | 0.00056 | 0.0036 |
| PSO-PID | 0.000130 | 0.000210 | 0.00230 | 0.0096 |
| PSO-I-PD | 0.000190 | 0.000560 | 0.00980 | 0.0663 |
| FA-PID | 0.000350 | 0.000100 | 0.00450 | 0.0023 |
| FA-I-PD | 0.000040 | 0.000080 | 0.00470 | 0.0196 |
| TLBO-PID | 0.000055 | 0.000023 | 0.00230 | 0.0210 |
| TLBO-I-PD | 0.000070 | 0.000090 | 0.00120 | 0.0020 |

four various performance criteria are used ITSE, ISE, ITAE, and IAE. However, ITSE [19, 32, 36], ISE [25, 26] and ITAE [11, 13] are mostly used for AGC. For the comparison among various performance criteria Eq (2a)–(2d) are executed in Matlab and identified that ITSE provides minimum error as compared to others criteria which are show in Table 1. Hence, ITSE criteria is preferred as an objective function to tune the gains of controller. Further, for the sake of comparison three other methods such as PSO, TLBO and FA are used for tuning of I-PD/ PID controller. The convergence rate for different methods using ITSE criteria is depicted in Fig 3. The time response performance is evaluated by comparing with DE-PID, LUS-PID, LUS-TLBO-PID fuzzy based LUS-TLBO-PID controllers. The parameters of FDO is chosen from Table 2. For simulation a population of 30 and generation of 60 numbers are considered. The optimization was performed by 30 times for each algorithm and the best optimal gains are picked during optimization which are specified in the Tables 3–6 for reheat thermal unit, hydro, gas and multi-source unit respectively.

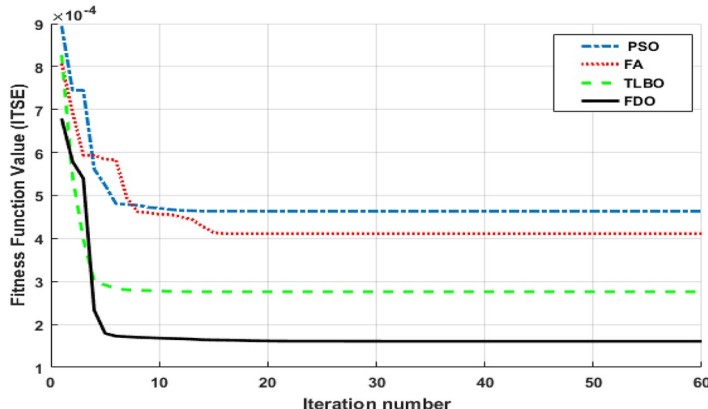

**Fig 3. Rate of convergence for different algorithms.**

**Table 2. Values of Fitness Dependent Optimizer parameters.**

| Parameters | Values | Parameters | Values | Parameters | Values |
|---|---|---|---|---|---|
| Number of Population $N_P$ | 30 | Number of generation $N_g$ | 60 | Lower bound $L_b$ | -2 |
| Upper bound $U_b$ | 2 | Number of dimensions $N_d$ | 9 | Weight Factor $\gamma$ | 0.0 |
| random number $\alpha$ | [-1, 1] | | - | - | - |

**Table 3. Optimum gains of PID /I-PD controllers optimized with different methods for reheat thermal unit.**

| Controller with Techniques | Reheat Thermal Power | | |
|---|---|---|---|
| | $K_p$ | $K_i$ | $K_d$ |
| FDO-PID | 0.260 | 1.903 | -1.680 |
| FDO-I-PD | 0.340 | 1.000 | -0.670 |
| PSO-PID | 1.023 | 1.010 | -0.223 |
| PSO-I-PD | 1.305 | 0.011 | -0.435 |
| FA-PID | 0.350 | 0.001 | 0.450 |
| FA-I-PD | -0.030 | 0.600 | 2.800 |
| TLBO-PID | 0.450 | 1.230 | -0.080 |
| TLBO-I-PD | 1.700 | 0.900 | -0.203 |

**Table 4. Optimum gains of PID /I-PD controllers optimized with different methods for hydro power unit.**

| Controller with Techniques | Hydro Power system | | |
|---|---|---|---|
| | $K_p$ | $K_i$ | $K_d$ |
| FDO-PID | 1.163 | 1.962 | -0.003 |
| FDO-I-PD | 1.200 | 1.012 | 1.001 |
| PSO-PID | 0.962 | 0.003 | 0.002 |
| PSO-I-PD | 1.012 | 1.001 | 1.012 |
| FA-PID | 1.002 | 1.002 | 0.001 |
| FA-I-PD | 1.013 | 1.012 | 0.080 |
| TLBO-PID | 0.450 | 0.093 | -0.080 |
| TLBO-I-PD | 0.700 | 0.010 | -0.230 |

### 3.0.1 Two-area reheat thermal power system

The Transfer Function (TF) model of thermal reheat power with a two-area interconnected system is provided in Fig 4. $R_{th1}$ and $R_{th2}$ denote the droop constant of area 1 and area 2 respectively, whereas $\Delta P_D$ indicates the change in load perturbation and $K_t$ represent thermal constant. Similarly, $\Delta F_1$ and $\Delta F_2$ represent the change of frequency in area 1 and area 2 respectively. The transfer function of the governor, reheat, and turbine for area 1 and area 2 are

**Table 5. Optimum gains of PID /I-PD controllers optimized with different methods for gas generation unit.**

| Controller with Techniques | Gas Power system | | |
|---|---|---|---|
| | $K_p$ | $K_i$ | $K_d$ |
| FDO-PID | 0.723 | 0.112 | -1.012 |
| FDO-I-PD | 0.305 | 0.811 | 0.029 |
| PSO-PID | 0.962 | 0.003 | 0.002 |
| PSO-I-PD | 1.060 | 0.723 | 0.687 |
| FA-PID | 1.002 | 1.002 | 0.001 |
| FA-I-PD | 0.062 | 1.100 | 0.458 |
| TLBO-PID | 1.072 | 1.007 | -0.238 |
| TLBO-I-PD | 0.723 | 0.112 | -1.012 |

**Table 6. Optimum values of I-PD/ PID controllers tuned with various techniques for power system.**

| Controller with Techniques | Reheat Thermal Power | | | Hydro Power System | | | Gas Power System | | |
|---|---|---|---|---|---|---|---|---|---|
| | $K_p$ | $K_i$ | $K_d$ | $K_p$ | $K_i$ | $K_d$ | $K_p$ | $K_i$ | $K_d$ |
| **FDO-PID** | 0.36 | **0.23** | **-0.67** | **0.47** | **0.23** | **-0.32** | **1.36** | **0.23** | **0.67** |
| **FDO-I-PD** | **0.56** | **0.09** | **-0.56** | **1.03** | **0.90** | **0.01** | **1.56** | **0.90** | **0.56** |
| PSO-PID | 0.23 | 1.00 | -0.03 | 1.96 | 1.10 | 0.01 | 0.23 | 1.00 | 0.23 |
| PSO-I-PD | 1.19 | 1.56 | 0.23 | 1.63 | 0.90 | 0.01 | 1.19 | 0.56 | -0.23 |
| FA-PID | 0.35 | 0.01 | -0.45 | 1.10 | 2.00 | 1.00 | 0.35 | 0.01 | 0.45 |
| FA-I-PD | 0.60 | 1.80 | 0.45 | 1.16 | 1.96 | -0.03 | 0.60 | -1.80 | 0.45 |
| TLBO-PID | 0.45 | 0.23 | 0.89 | 1.07 | 1.90 | 0.10 | 0.45 | 0.23 | -0.23 |
| TLBO-I-PD | 0.70 | 1.90 | 0.49 | 2.00 | 0.90 | 0.10 | 0.70 | 1.90 | -0.23 |

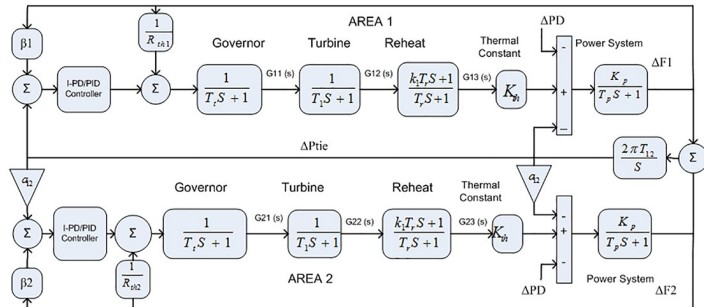

**Fig 4. Two-area model with reheat thermal power system.**

represented in below equations respectively.

$$G_{11}(S) = G_{21}(S) = \frac{1}{T_t s + 1} \tag{6}$$

$$G_{12}(S) = G_{22}(S) = \frac{K_1 T_r s + 1}{T_r s + 1} \tag{7}$$

$$G_{13}(S) = G_{23}(S) = \frac{1}{T_1 s + 1} \tag{8}$$

So, the overall transfer function for reheat thermal power system for area-1 and area-2 is given in Eq (9).

$$G_1(S) = G_2(S) = \frac{(K_1 T_r s + 1)}{(T_t s + 1)(T_r s + 1)(T_1 s + 1)} \tag{9}$$

The TF diagram for two-area reheat thermal IPS are developed in Matlab/Simulink using the values of the gains from Table 7. The system is evaluated with I-PD/ PID controllers with 1% SLP at $t = 0s$. Figs (5–10) depicts the responses of load frequency in each area and tie-line power. In pursuance to examine the effectiveness of the FDO-PID and FDO-I-PD control strategies, the output responses are compared with several other algorithms based PID and

**Table 7. Parameter setting for two-area interconnected power system [13].**

| Parameters | Values | Parameters | Values | Parameters | Values |
|---|---|---|---|---|---|
| $\beta_1, \beta_2$ | 0.431 MW/Hz | $R_{th1}, R_{th2}, R_{hy1}, R_{hy2}, R_{g1}, R_{g2}$ | 2.40 Hz/p.u | $T_{gh}$ | 0.080 s |
| $T_t$ | 0.30 s | $K_1$ | 0.30 | $T_r$ | 10 s |
| $K_P$ | 68.95 | $T_p$ | 11.490 s | $T_{12}$ | 0.0430 |
| $a_{12}$ | -1 | $T_w, y_c$ | 1 s | $T_{rs}$ | 5 s |
| $T_{rh}$ | 28.70 s | $T_{gh}, T_{cd}, T_{DC}$ | 0.20 s | $x_c$ | 0.60 s |
| $K_g$ | 0.1304 | $K_{DC}, x_g$ | 1 | $b_g$ | 0.050 s |
| $K_t$ | 0.5434 | $T_{cr}$ | 0.010 s | $T_F$ | 0.230 s |
| $K_h$ | 0.3268 | - | - | - | - |

I-PD controllers including PSO-PID, PSO-I-PD, FA-PID, FA-I-PD, TLBO-PID, and TLBO-I-PD. The comparison of transient response specifications are quantified and presented in Table 8.

The comparison of results from Fig 5 and Table 8 reveal that FDO-PID controller completely eliminates overshoot $O_{sh}$ as compared to PSO, FA and TLBO based PID controller which is dire need of a controller for the system stability. Further, settling time ($T_s$) and under-shoot ($U_{sh}$) yielded by FDO-PID are better than PSO/FA/TLBO based PID controller. The results shown in Fig 6 further reveal that FDO-PID controller provides less $O_{sh}$ than PSO-PID however, at the cost of slight increase in $T_s$, but better than FA-PID and TLBO-PID.

The results given in Figs 7 and 8 further shows that FDO-I-PD controller for both area 1 and area 2 outperform in terms of $T_s$, $O_{sh}$ and $U_{sh}$ which is less than PSO/FA/TLBO base tuned PID controllers. The results shown in Fig 9 reveals that FDO-PID has lesser $O_{sh}$ as compared to other controllers at the cost of increase in settling time for PSO-PID and FA-PID but still better than TLBO-PID. The outcome given in Fig 10 express the overall superiority of FDO-I-PD than other controllers in all aspects i.e less $T_s$, $O_{sh}$ and $U_{sh}$. The overall comparison of the results for area 1 and 2 for reheat thermal power system is quantified and presented in Table 8.

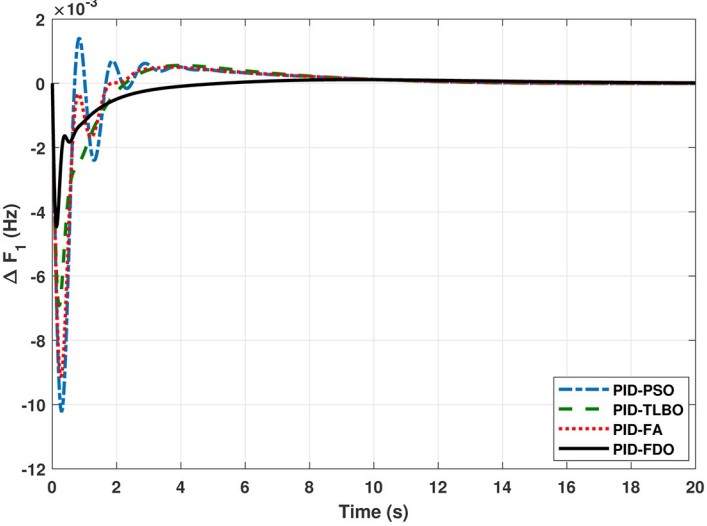

**Fig 5. Results for reheat thermal unit in area 1 with PID controller.**

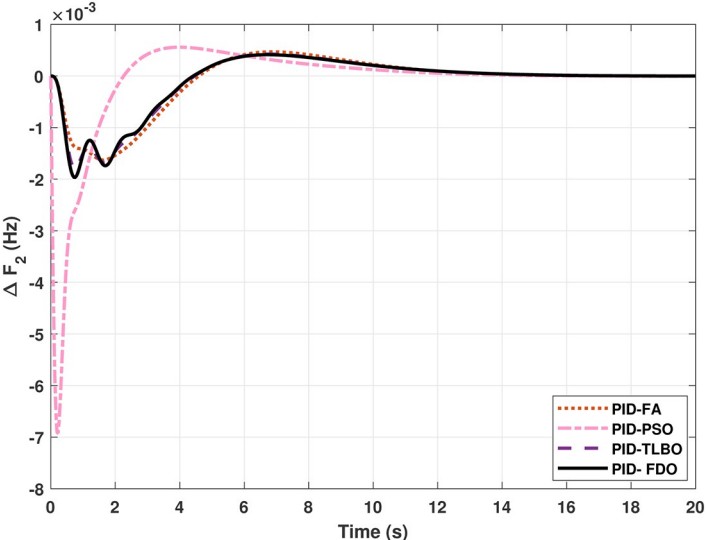

**Fig 6. Results for reheat thermal unit in area 2 with PID controller.**

**3.1 Two area hydro power system.** The transfer function (TF) diagram for the hydro power system with two areas are shown in Fig 11. Where $K_h$ represent the hydro constant of the system and $R_{h1}$ and $R_{h2}$ shows droop constant of the hydro unit for area 1 and 2 respectively. The transfer function of the governor, pen stock turbine, and transient droop compensation for area 1 and area 2 are represented in below equations respectively.

$$G_{11}(S) = G_{21}(S) = \frac{1}{T_{gh}s + 1} \tag{10}$$

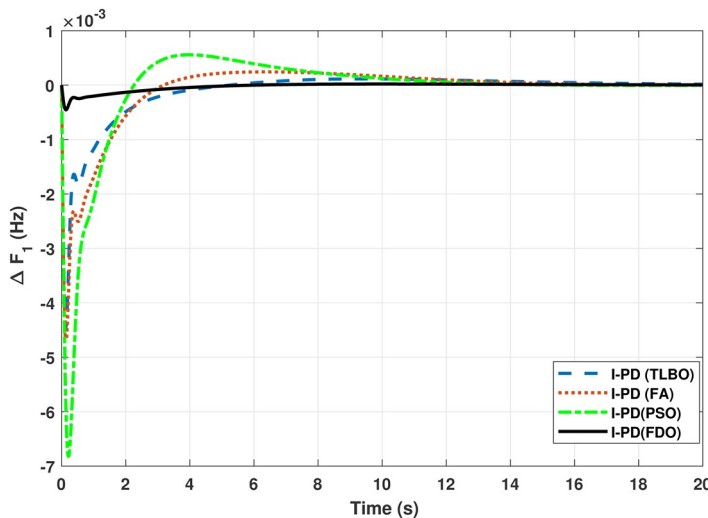

**Fig 7. Results for reheat thermal unit in area 1 with I-PD controller.**

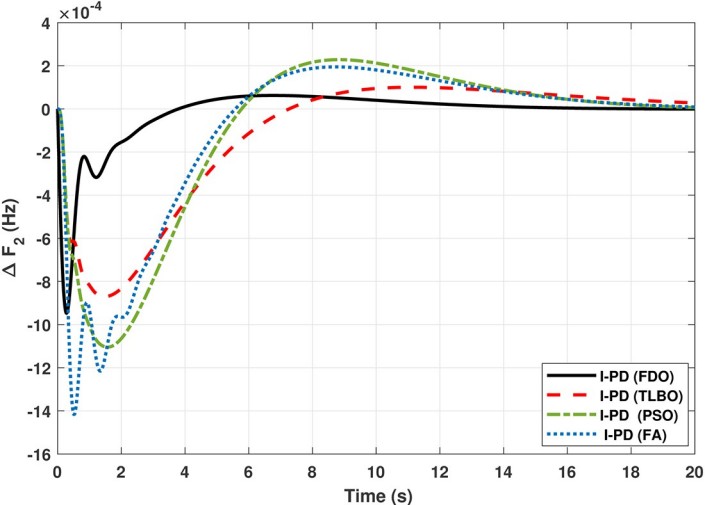

**Fig 8. Results for reheat thermal unit in area 2 with I-PD controller.**

$$G_{12}(S) = G_{22}(S) = \frac{T_r s + 1}{T_{rh} s + 1} \tag{11}$$

$$G_{13}(S) = G_{23}(S) = \frac{-T_w s + 1}{0.5 T_w s + 1} \tag{12}$$

Hence, the overall transfer function for hydro power system for area-1 and area-2 is given in

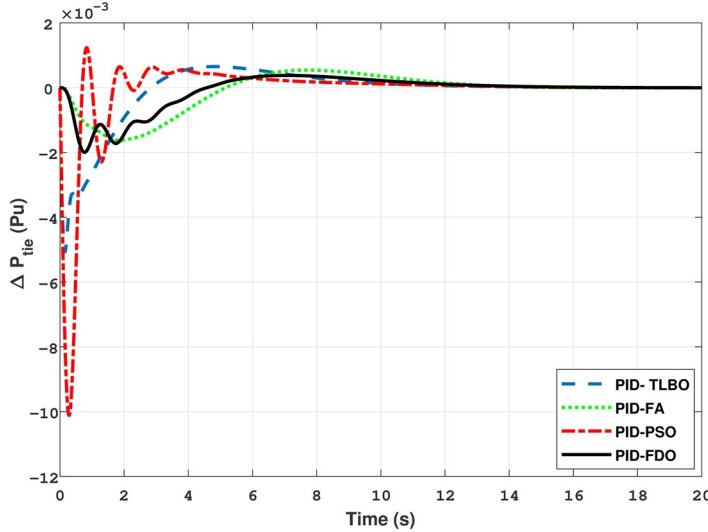

**Fig 9. Results for reheat thermal unit of tie-line power with PID controller.**

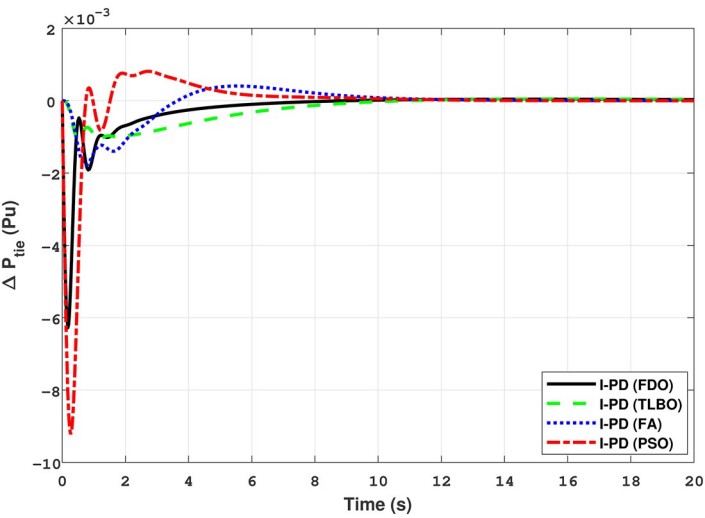

**Fig 10. Results for reheat thermal unit of tie-line power with I-PD controller.**

**Table 8. Comparative performance between different algorithms for reheat thermal with two-area power system.**

| Controller with Techniques | Settling Time $T_s$ s | | | Overshoot $O_{sh}$ | | | Undershoot $U_{sh}$ | | |
|---|---|---|---|---|---|---|---|---|---|
| | $\Delta F_1$ | $\Delta F_2$ | $\Delta P_{tie}$ | $\Delta F_1$ | $\Delta F_2$ | $\Delta P_{tie}$ | $\Delta F_1$ | $\Delta F_2$ | $\Delta P_{tie}$ |
| **FDO-PID** | **2.9** | **9.2** | **10.3** | **0.00000** | **0.00250** | **0.00021** | **-0.00400** | **-0.00200** | **-0.00200** |
| **FDO-I-PD** | **1.9** | **10.1** | **4.9** | **0.00000** | **0.00012** | **0.00000** | **-0.00050** | **-0.00090** | **-0.00620** |
| PSO-PID | 6.5 | 9.1 | 8.4 | 0.00180 | 0.01630 | 0.00160 | -0.01000 | -0.00690 | -0.01010 |
| PSO-I-PD | 8.2 | 14.2 | 6.2 | 0.00150 | 0.00030 | 0.00930 | -0.00680 | -0.00130 | -0.00910 |
| FA-PID | 6.3 | 10.4 | 8.7 | 0.01200 | 0.00280 | 0.00081 | -0.08400 | -0.01700 | -0.00500 |
| FA-I-PD | 10.9 | 16.5 | 9.3 | 0.00072 | 0.00025 | 0.00710 | -0.00480 | -0.00110 | -0.00180 |
| TLBO-PID | 6.4 | 10.2 | 11.6 | 0.01300 | 0.00270 | 0.00091 | -0.06300 | -0.01800 | -0.00160 |
| TLBO-I-PD | 3.8 | 14.3 | 8.3 | 0.00020 | 0.00310 | 0.00000 | -0.00460 | -0.00070 | -0.01100 |

Eq (13).

$$G_1(S) = G_2(S) = \frac{(T_r s + 1)(-T_w s + 1)}{(T_{gh} s + 1)(T_{rh} s + 1)(0.5 T_w s + 1)}$$

(13)

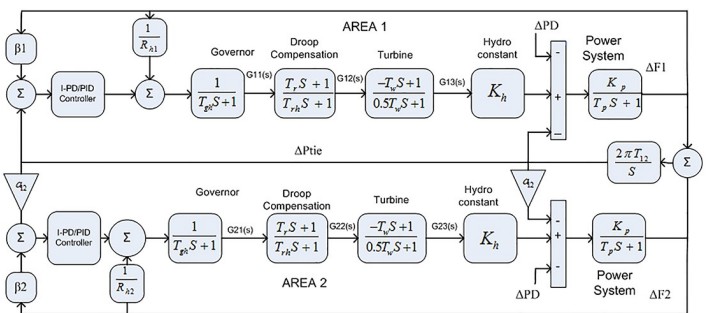

**Fig 11. Two-area model with hydro power unit.**

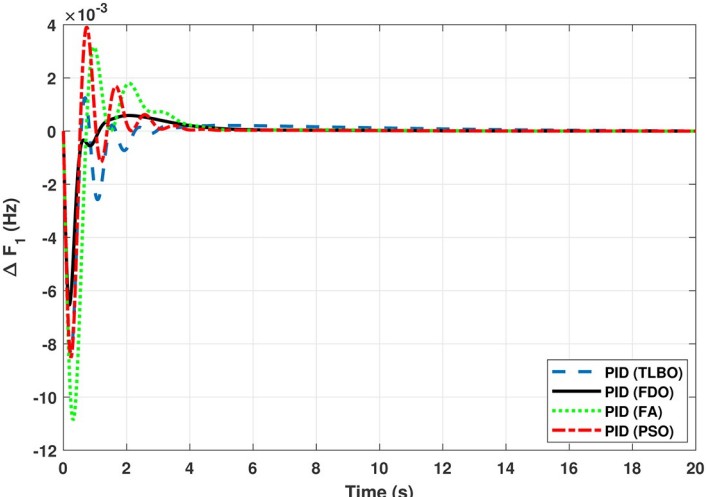

**Fig 12. Results for hydro power unit in area 1 with PID controller.**

The TF model of two-area hydro power system have been assessed with FDO-PID and FDO-I-PD. The results are compared with PID and I-PD with other tuning techniques i.e. PSO, TLBO and FA. The results obtained from two-area hydro power system with PID and I-PD controllers are given in Figs (12–17).

The results obtained from interconnected hydro power system with PID controller are given in Fig 12 which shows the better performance of FDO-PID controller as compared to TLBO/FA/PSO based PID controller. The results in Fig 13 reflect that FDO-PID provides output response with zero overshoot ($O_{sh}$ = 0.0000) as compared to PSO-PID ($O_{sh}$ = 0.0029), FA-PID ($O_{sh}$ = 0.0019) and TLBO-PID ($O_{sh}$ = 0.0006) controllers. Similarly, the results also express that FDO-PID yields lesser $U_{sh}$ and $T_s$ as compared to PID controller with other tuning techniques. The results depicted in Fig 14 indicate that FDO-I-PD controller provides

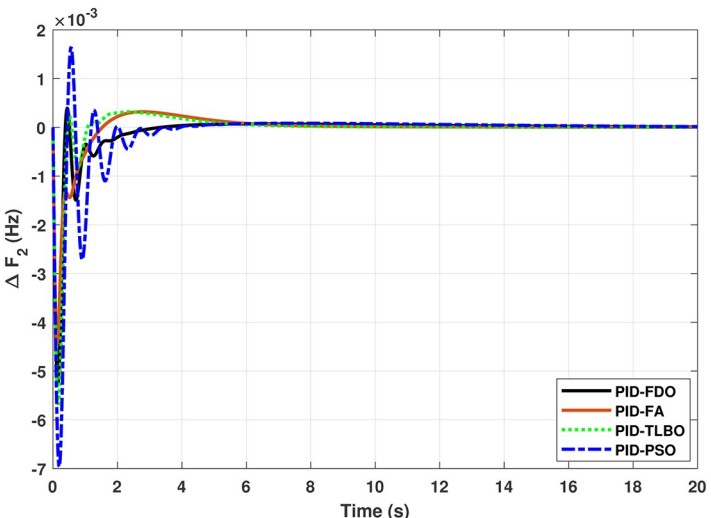

**Fig 13. Results for hydro power unit in area 2 with PID controller.**

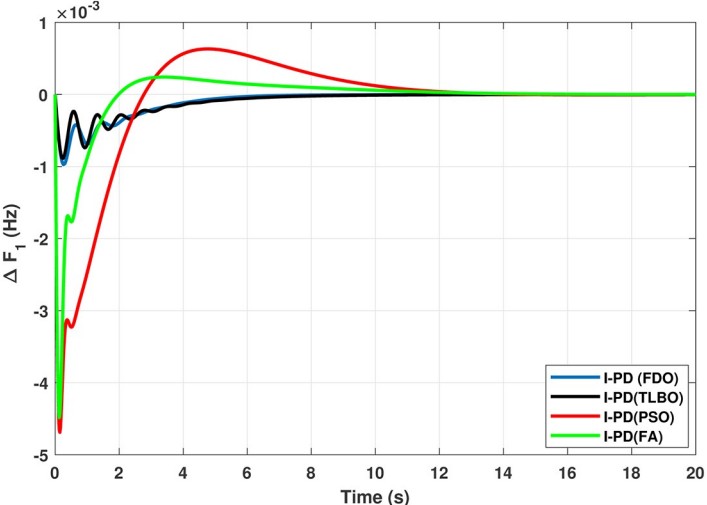

**Fig 14. Results for hydro power unit in area 1 with I-PD controller.**

excellent results with zero $O_{sh}$ in load frequency of area 1, less settling time and undershoot as associated with other techniques like PSO, FA and TLBO with same controllers. The results illustrated in Fig 15 express that FDO base tuned PID controller have less settling time, undershoot and overshoot than PSO, TLBO and FA base tuned PID controller. However, a minor change of 0.001 in overshoot can be clearly seen than PID based controller optimized with FDO algorithm. From Fig 16 it can be seen that FDO base tuned algorithm have less settling time nonetheless of minor increased in overshoot which is completely eliminated by FDO based tuned I-PD controller which is indicated in Fig 17 but still better than other used techniques in terms of $T_s$, $O_{sh}$ and $U_{sh}$. A comprehensive comparison results for two-area hydro power system and change in tie-lines in terms of $T_s$, $O_{sh}$ and $U_{sh}$ are given in Table 9. The representation of bold values indicates the best results.

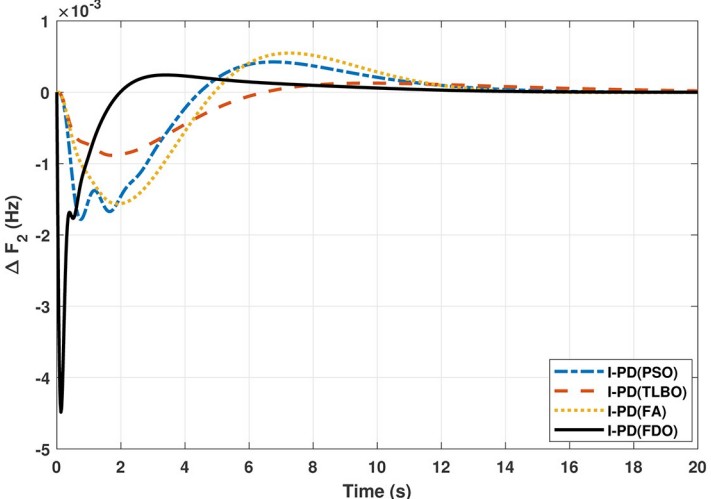

**Fig 15. Results for hydro power unit in area 2 with I-PD controller.**

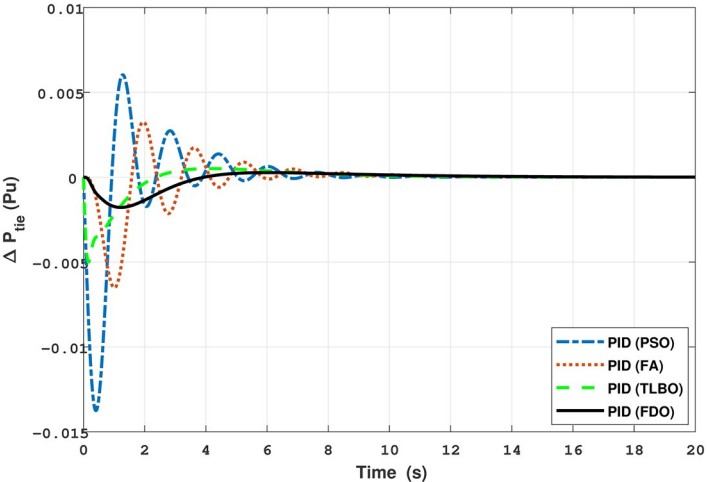

**Fig 16. Results for hydro power unit of tie-line power with PID controller.**

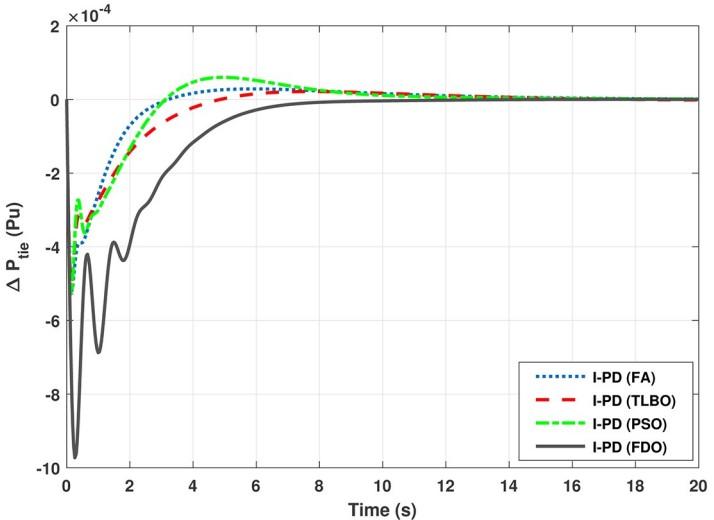

**Fig 17. Results for hydro power unit of tie-line power with I-PD controller.**

**Table 9. Comparative performance between different algorithms for two-area hydro power system.**

| Controller with Techniques | Settling Time $T_s$ | | | Overshoot $O_{sh}$ | | | Undershoot $U_{sh}$ | | |
|---|---|---|---|---|---|---|---|---|---|
| | $\Delta F_1$ | $\Delta F_2$ | $\Delta P_{tie}$ | $\Delta F_1$ | $\Delta F_2$ | $\Delta P_{tie}$ | $\Delta F_1$ | $\Delta F_2$ | $\Delta P_{tie}$ |
| **FDO-PID** | **2.83** | **2.30** | **3.20** | **0.00000** | **0.00000** | **0.00020** | **-0.00760** | **-0.00430** | **-0.00023** |
| **FDO-I-PD** | **2.90** | **5.30** | **4.00** | **0.00000** | **0.00100** | **0.00000** | **-0.00410** | **-0.00190** | **-0.00100** |
| PSO-PID | 7.30 | 6.40 | 10.10 | 0.00290 | 0.00190 | 0.00600 | -0.06400 | -0.00520 | -0.01300 |
| PSO-I-PD | 6.70 | 9.80 | 8.70 | 0.00600 | 0.00500 | 0.00730 | -0.00420 | -0.00130 | -0.00460 |
| FA-PID | 4.30 | 5.60 | 10.20 | 0.00190 | 0.00060 | 0.00420 | -0.04300 | -0.00700 | -0.00600 |
| FA-I-PD | 5.40 | 10.1 | 6.80 | 0.00800 | 0.00700 | 0.00490 | -0.00930 | -0.00150 | -0.00430 |
| TLBO-PID | 2.40 | 4.20 | 4.60 | 0.00060 | 0.00056 | 0.00100 | -0.06200 | -0.00560 | -0.00500 |
| TLBO-I-PD | 3.10 | 6.20 | 4.40 | 0.00210 | 0.00300 | 0.00000 | -0.00420 | -0.00170 | -0.00900 |

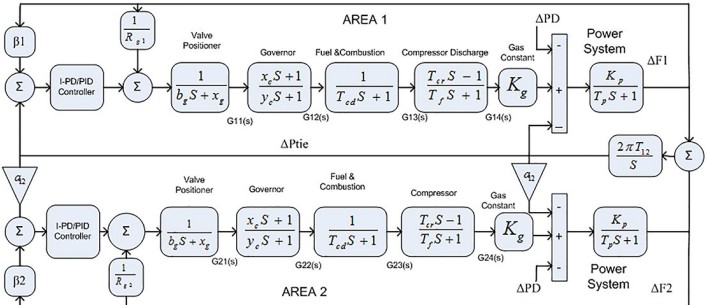

**Fig 18. Two-area model with gas power system.**

## 3.2 Two-area gas power system

The Transfer Function (TF) diagram of two area gas power generation is show in Fig 18. Where $R_{g1}$ and $R_{g2}$ represents droop constant of the gas unit for area 1 and 2 respectively and $K_g$ represent the gas constant. The transfer function of the valve position, speed governor, compressor discharge and fuel with combustion reaction for area 1 and area 2 are represented in below equations respectively.

$$G_{11}(S) = G_{21}(S) = \frac{1}{b_g s + x_g} \tag{14}$$

$$G_{12}(S) = G_{22}(S) = \frac{x_c s + 1}{g_c s + 1} \tag{15}$$

$$G_{13}(S) = G_{23}(S) = \frac{1}{T_{cd} s + 1} \tag{16}$$

$$G_{14}(S) = G_{24}(S) = \frac{T_{cr} s - 1}{T_f s + 1} \tag{17}$$

Hence, the overall transfer function for gas power system for area-1 and area-2 is given in Eq (18).

$$G_1(S) = G_2(S) = \frac{(x_c s + 1)(T_{cr} s - 1)}{(b_g s + x_g)(g_c s + 1)(T_{cd} s + 1)(T_f s + 1)} \tag{18}$$

The TF model of two-area gas power system are established in Matlab/Simulink using Table 7. The system is assessed with FDO base optimized PID/I-PD controllers to evaluate the achievement of the proposed techniques and their outcomes are compared with some recent optimization algorithms like TLBO, FA and PSO. The results obtained from two-area gas power system for area 1 and 2 with PID and I-PD controllers are given in Figs (19–24).

In Fig 19 the results reveal that FDO base PID controller completely eliminate the overshoot ($o_{sh}$ = 0.000), less settling time ($T_s$ = 2.1) and undershoot ($U_{sh}$ = −0.0048) as compared to PSO ($O_{sh}$ = 0.0036, $T_s$ = 5.9s, $U_{sh}$ = −0.0072), FA ($O_{sh}$ = 0.021, $T_s$ = 3.9s, $U_{sh}$ = −0.0083), and TLBO ($O_{sh}$ = 0.019, $T_s$ = 5.2s, $U_{sh}$ = −0.0049) base PID controller. The results depicted in Fig 20 express that FDO base tuned PID controller have similar settling time and overshoot but less

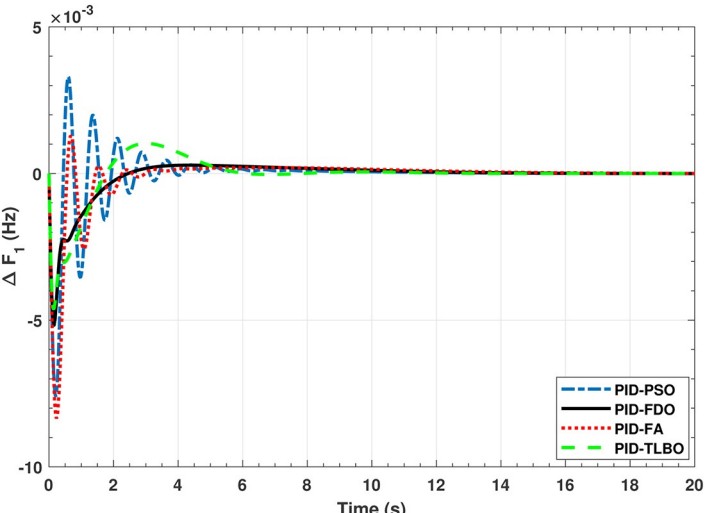

**Fig 19. Results for gas unit in area 1 with PID controller.**

undershoot than TLBO base tuned PID controller, nonetheless better than remaining techniques i.e. FA and PSO.

The results given in Figs 21 and 22 indicates that FDO base tuned I-PD controller for change in frequency with area 1 and area 2 have good performance in terms of $T_s$, $O_{sh}$ and $U_{sh}$ which is less than PSO/FA/TLBO base tuned PID controllers. In inclusive comparison of the results for area 1 and 2 of gas power system in settling time, overshoot, undershoot and different performance indices are given in Table 10. The outcome shown in Fig 23 articulates that FDO base tuned PID controller for change in tie- line power have better results in all aspects i.e $T_S$, $O_{sh}$ and $U_{sh}$ from other applied techniques with same controller. From Fig 24 it can observed that

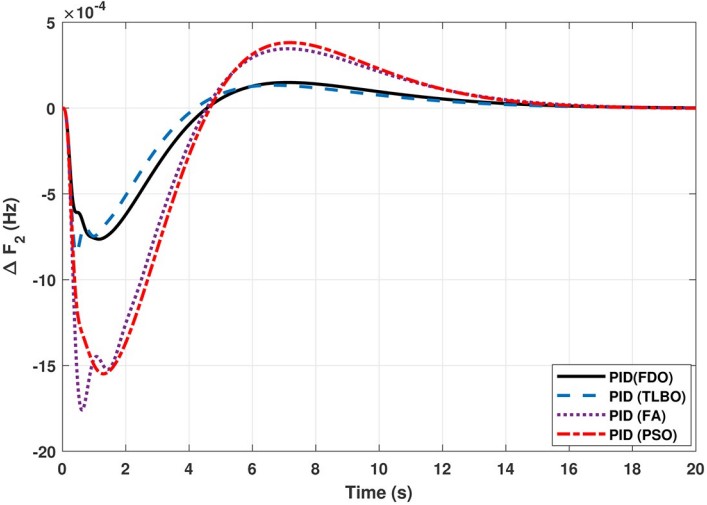

**Fig 20. Results for gas unit in area 2 with PID controller.**

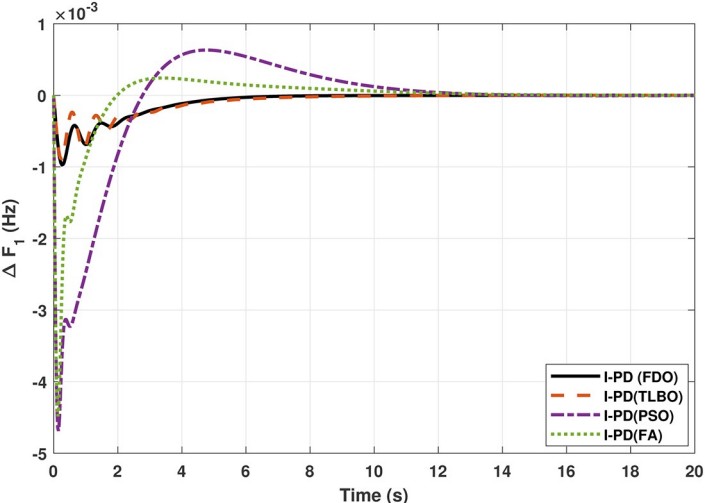

**Fig 21. Results for gas unit in area 1 with I-PD controller.**

FDO base optimized I-PD controller have better result than same controller tuned with PSO/FA/TLBO in terms of lesser $O_{sh}$, $T_S$ and $U_{Sh}$. The best results are indicated with bold formats.

## 3.3 Two-area multi-source interconnected power system

The Transfer Function (TF) diagram for multi-source with two-area IPS is depicted in Fig 25. In each control area the system is equipped with reheat thermal, gas power and hydro power unit which makes the system more complicated as compared to generation unit which are individually applied in two area power system. Two-area multi-source interconnected power system have been developed in Matlab Simulink and the values have been used from Table 7.

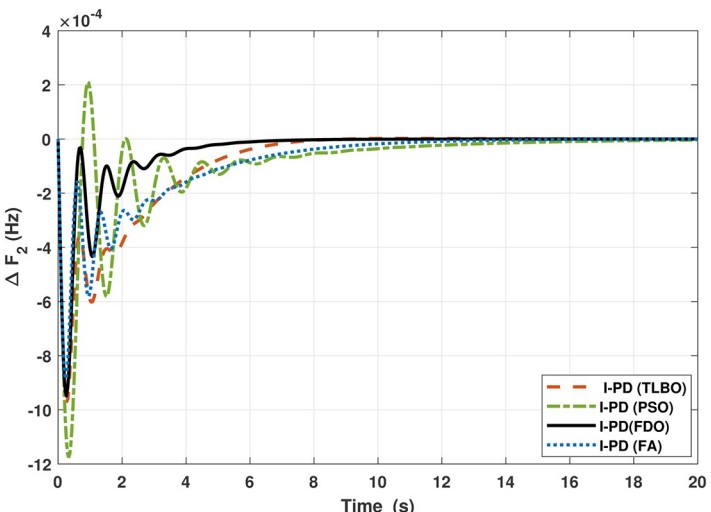

**Fig 22. Results for gas unit in area 2 with I-PD controller.**

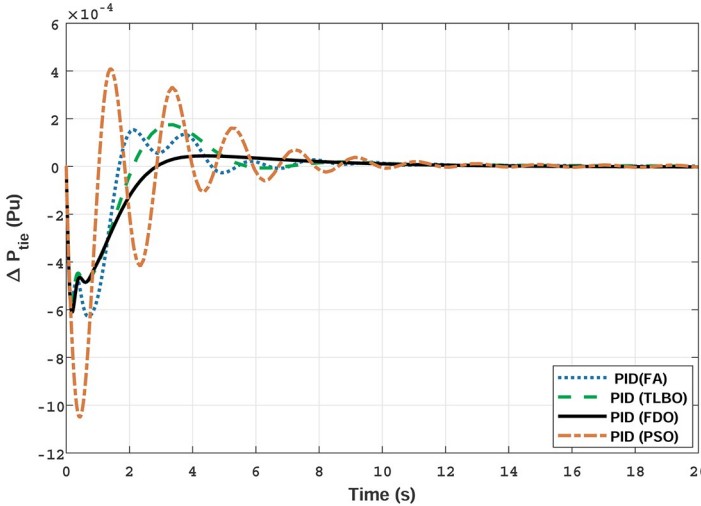

**Fig 23. Results for gas unit of tie-line power with PID controller.**

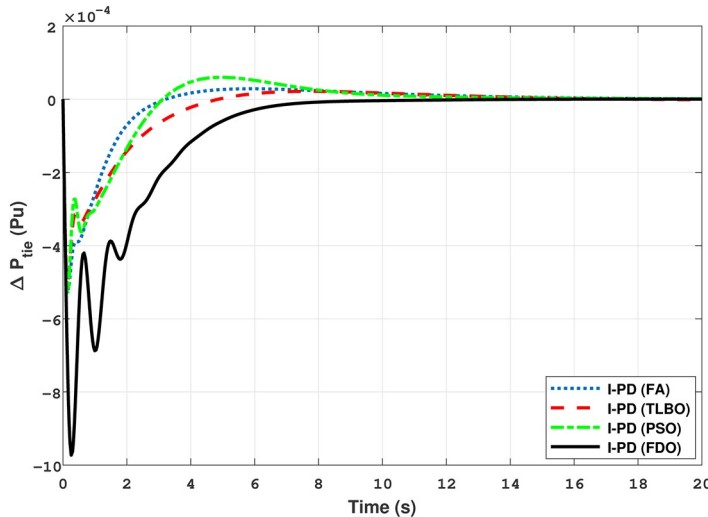

**Fig 24. Results for gas unit of tie-line power with I-PD controller.**

**Table 10. Comparative performance between different algorithms for two-area gas power system.**

| Controller with Techniques | Settling Time $T_s$ | | | Overshoot $O_{sh}$ | | | Undershoot $U_{sh}$ | | |
|---|---|---|---|---|---|---|---|---|---|
| | $\Delta F_1$ | $\Delta F_2$ | $\Delta P_{tie}$ | $\Delta F_1$ | $\Delta F_2$ | $\Delta P_{tie}$ | $\Delta F_1$ | $\Delta F_2$ | $\Delta P_{tie}$ |
| **FDO-PID** | **2.1** | **10.2** | **3.1** | **0.00000** | **0.00013** | **0.00003** | **0.00730** | **-0.00720** | **-0.00059** |
| **FDO-I-PD** | **4.0** | **3.9** | **5.2** | **0.00000** | **0.00000** | **0.00000** | **-0.00100** | **-0.00084** | **-0.00047** |
| PSO-PID | 5.9 | 12.4 | 10.4 | 0.03600 | 0.00390 | 0.00040 | -0.00480 | -0.00160 | -0.00110 |
| PSO-I-PD | 8.7 | 9.6 | 5.9 | 0.00730 | 0.00201 | 0.00930 | -0.00460 | -0.00019 | -0.00054 |
| FA-PID | 3.9 | 12.3 | 7.9 | 0.00210 | 0.00320 | 0.00016 | -0.00490 | -0.00170 | -0.00064 |
| FA-I-PD | 6.8 | 9.2 | 6.3 | 0.00490 | 0.00000 | 0.00021 | -0.00430 | -0.00082 | -0.00052 |
| TLBO-PID | 5.2 | 10.2 | 6.8 | 0.01000 | 0.00013 | 0.00017 | -0.00830 | -0.00080 | -0.00061 |
| TLBO-I-PD | 4.4 | 5.9 | 10.1 | 0.00000 | 0.00000 | 0.00000 | -0.00900 | -0.00089 | -0.00096 |

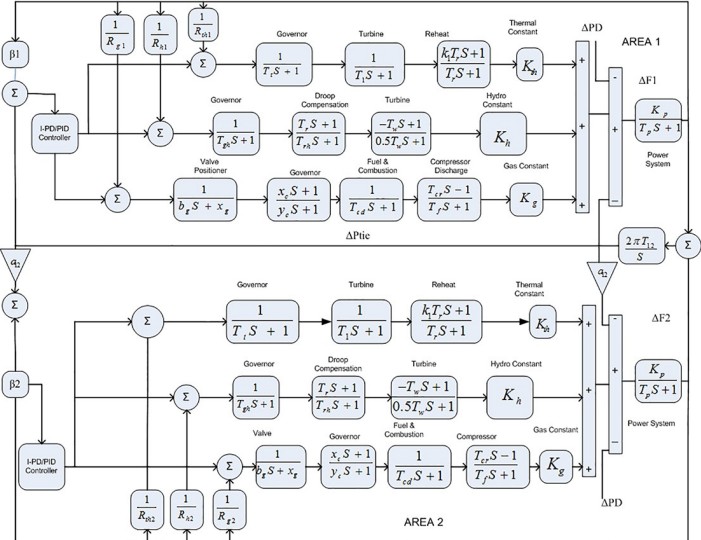

**Fig 25. Two-area with multi-source power system.**

The system is assessed with FDO base tuned I-PD/ PID controllers and the superiority of the proposed techniques are compared with other techniques i.e LUS-PID, LUS-TLBO-PID, FA-PID, TLBO-PID, PSO-PID and DE-PID. The results are depicted in Figs (26)–(31).

The results shown in Fig 26 for multi-source interconnected power system of area 1 reveals that PID controller with FDO base algorithm have no overshoot, less settling time and undershoot as associated with other techniques like PSO, TLBO and FA. The results depicted in Fig 27 express that FDO base tuned PID controller have less overshoot and settling time than

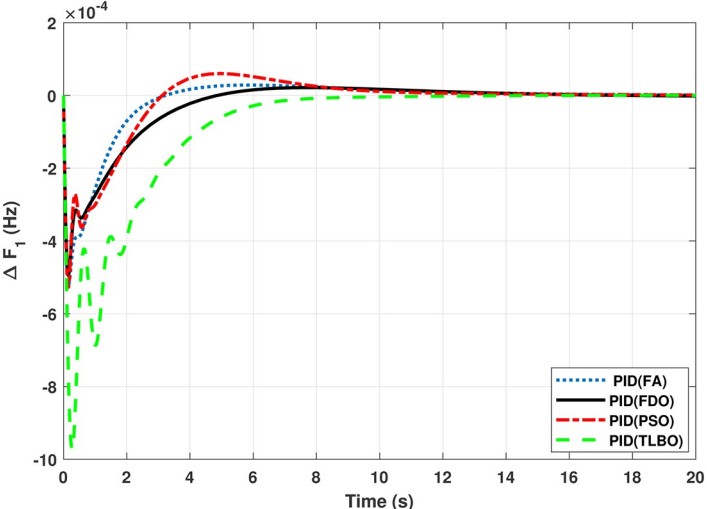

**Fig 26. Results for multi-source in area 1 with PID controller.**

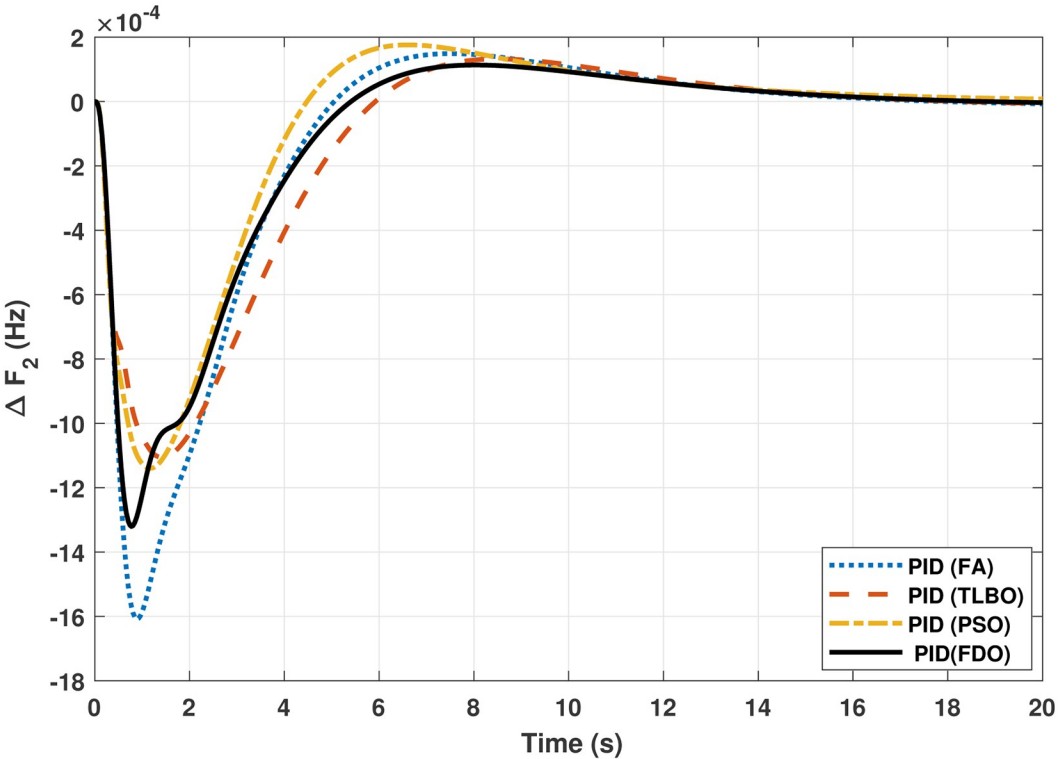

**Fig 27. Results for multi-source in area 2 with PID controller.**

TLBO and FA base tuned PID controller however at the cost of 0.001 increase in undershoot, but shows better improvement than PSO base tuned PID controller.

The results shown in Fig 28 express that FDO-PID has less settling time $T_s$, undershoot $U_s h$ and overshoot $O_s h$ than PSO-PID, TLBO-PID and FA-PID. However, a minor change of

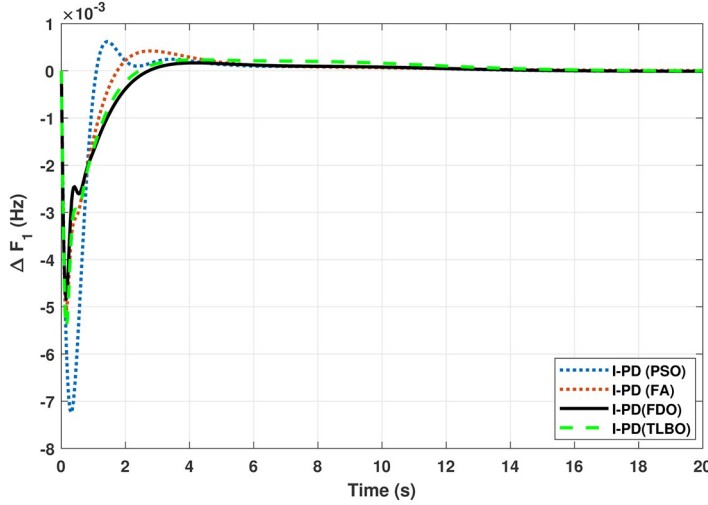

**Fig 28. Results for multi-source in area 1 with I-PD controller.**

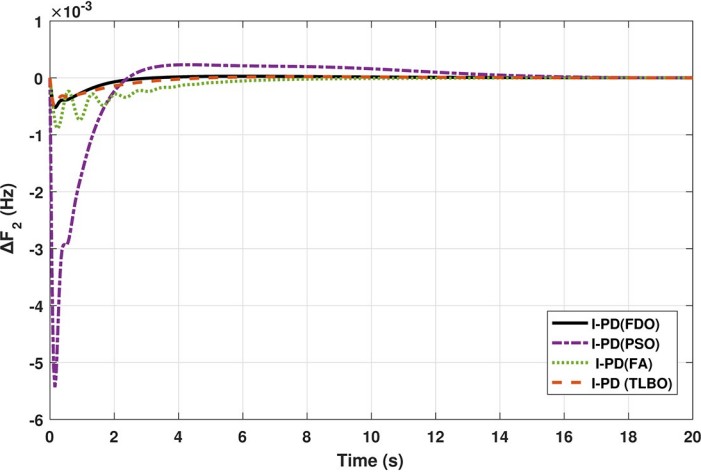

**Fig 29. Results for multi-source in area 2 with I-PD controller.**

0.001 in $O_s h$ can be clearly seen than FDO-PID. The results given in Fig 29 indicates that FDO-I-PD technique for change in frequency of area 2 has no overshoot, less settling time and undershoot as compared with other techniques like PSO-I-PD, FA-I-PD and TLBO-I-PD. The results given in Figs 30 and 31 reflect that FDO-I-PD for change in tie line power have good performance in terms of $T_s$, $O_{sh}$ and $U_{sh}$ which is less than PSO/FA/TLBO base optimized PID/I-PD controllers. A comprehensive comparison results for two-area multi- source power system and tie lines in terms of $T_s$, $O_{sh}$ and $U_{sh}$ are given in Table 11. The representation of bold values indicates the best results. the percentage improving comparing with different techniques is shown in bar chart of Fig 32.

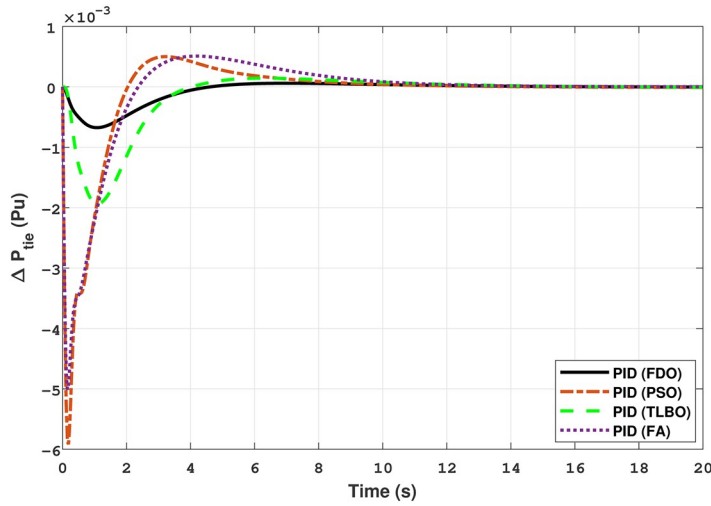

**Fig 30. Results for multi-source of tie-line power with PID controller.**

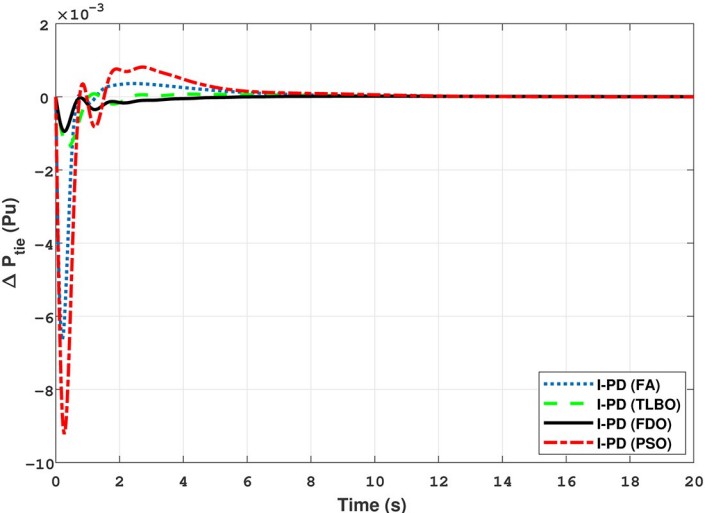

**Fig 31. Results for multi-source of tie-line power with I-PD controller.**

## 3.4 Sensitivity analysis

The robustness of an I-PD controller is verified by varying the system parameters of two areas multi-generation system within a range of ± 50% with a step of ± 25%. Fitness dependent algorithm (FDO) is employed to check the performance of the proposed controller by varying some parameters of the system from their nominal values such as turbine constant ($T_t$), droop constant $R$ and governor constant ($T_g$). The results yielded from sensitivity analysis for $\Delta F_1$, $\Delta F_2$, and $\Delta P_{tie}$ with variation in $T_t$, $R$ and $T_g$ is depicted in Figs 33–41. It can be observed from results that our proposed controller provides robustness by changing system parameters $T_t$, $R$ and $T_g$ for $\Delta F_1$, $\Delta F_2$, and $\Delta P_{tie}$ respectively within a range of ± 50%. Further, it can also be observed from Figs 33–41 that the system response plotted with variation in various

**Table 11. Comparative performance between different algorithms for two-area multi-source power system.**

| Controller with Techniques | Settling Time $T_s$ | | | Overshoot $O_{sh}$ | | | Undershoot $U_{sh}$ | | |
|---|---|---|---|---|---|---|---|---|---|
| | $\Delta F_1$ | $\Delta F_2$ | $\Delta P_{tie}$ | $\Delta F_1$ | $\Delta F_2$ | $\Delta P_{tie}$ | $\Delta F_1$ | $\Delta F_2$ | $\Delta P_{tie}$ |
| **FDO-PID** | 5.20 | 12.70 | 4.30 | 0.00000 | 0.00021 | 0.00020 | -0.00047 | -0.00130 | -0.00056 |
| **FDO-I-PD** | **2.30** | 1.65 | 2.10 | 0.00000 | 0.00000 | 0.00000 | -0.00450 | -0.00500 | -0.00058 |
| PSO-PID | 5.90 | 13.40 | 9.30 | 0.00930 | 0.00250 | 0.00630 | -0.00054 | -0.00016 | -0.00580 |
| PSO-I-PD | 2.90 | 9.20 | 6.80 | 0.00400 | 0.00400 | 0.00080 | -0.00720 | -0.05500 | -0.00910 |
| FA-PID | 6.30 | 13.10 | 7.10 | 0.00021 | 0.00036 | 0.00610 | -0.00052 | -0.00110 | -0.00510 |
| FA-I-PD | 4.30 | 4.10 | 5.90 | 0.00000 | 0.00000 | 0.00043 | -0.00490 | -0.00900 | -0.00640 |
| TLBO-PID [30] | 9.37 | 3.76 | 4.76 | 0.00172 | 0.00043 | 0.00017 | -0.01972 | -0.01279 | -0.00307 |
| TLBO-I-PD | **2.20** | 2.20 | 2.20 | 0.00000 | 0.00000 | 0.00011 | -0.00530 | -0.00500 | -0.00130 |
| LUS-TLBO Fuzzy-PID [30] | 5.26 | 2.96 | 2.36 | 0.00055 | 0.00021 | 0.00008 | -0.00895 | -0.00301 | -0.00096 |
| DE Fuzzy-PID [30] | 5.30 | 2.90 | 2.60 | 0.00073 | 0.00024 | 0.00009 | -0.01297 | -0.00589 | -0.00160 |
| LUS-TLBO-PID [30] | 9.22 | 3.34 | 4.98 | 0.00167 | 0.00042 | 0.00015 | -0.01689 | -0.01133 | -0.00289 |
| LUS-PID [30] | 9.29 | 4.62 | 6.36 | 0.00185 | 0.00046 | 0.00015 | -0.02105 | -0.01517 | -0.00358 |
| DE-PID [11] | 13.84 | 8.35 | 9.35 | 0.00203 | 0.00077 | 0.00019 | -0.02657 | -0.02214 | -0.00475 |

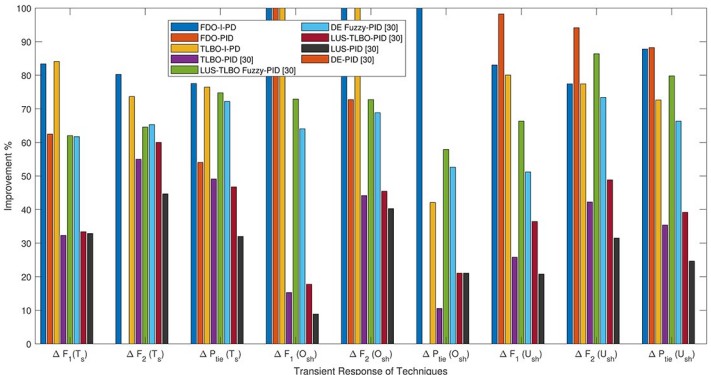

**Fig 32. Comparison in sense of improvement% with reference DE-PID [11].**

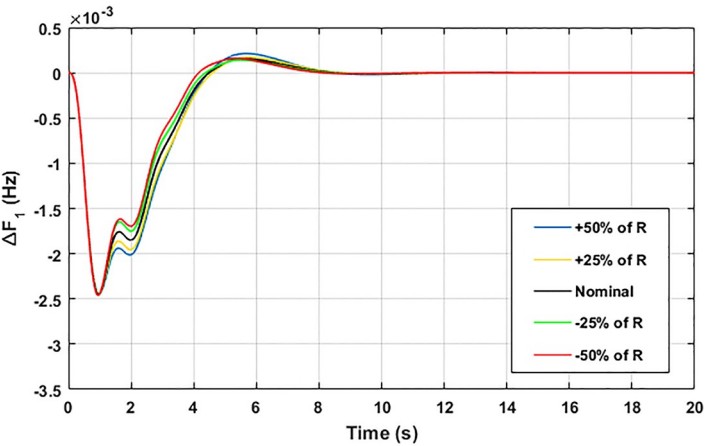

**Fig 33. Results for variation in $R$ with $\Delta F_1$.**

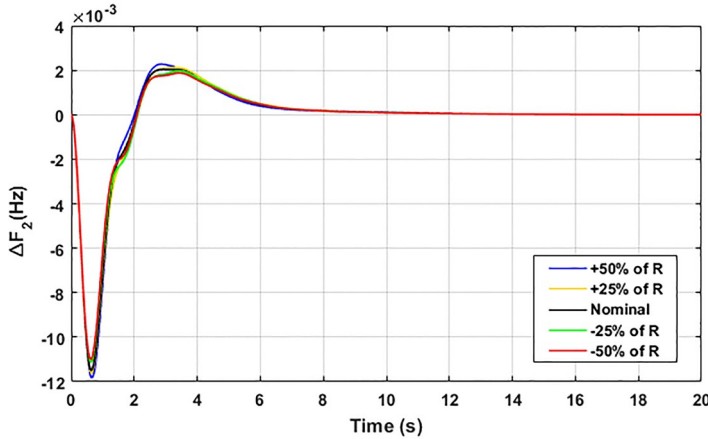

**Fig 34. Results for variation in $R$ with $\Delta F_2$.**

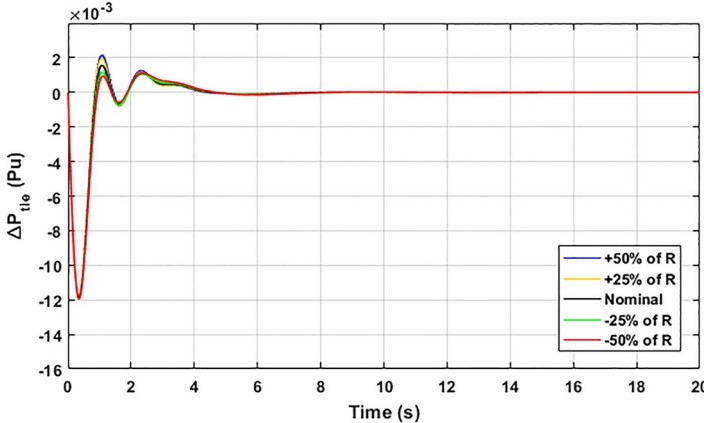

**Fig 35. Results for variation in *R* with $\Delta P_{tie}$.**

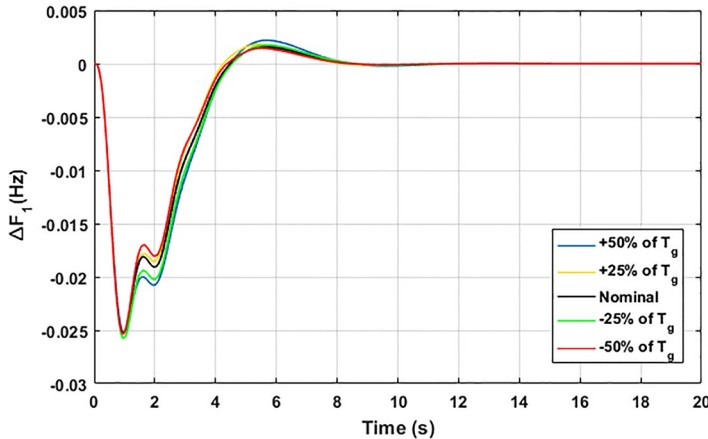

**Fig 36. Results for variation in $T_g$ with $\Delta F_1$.**

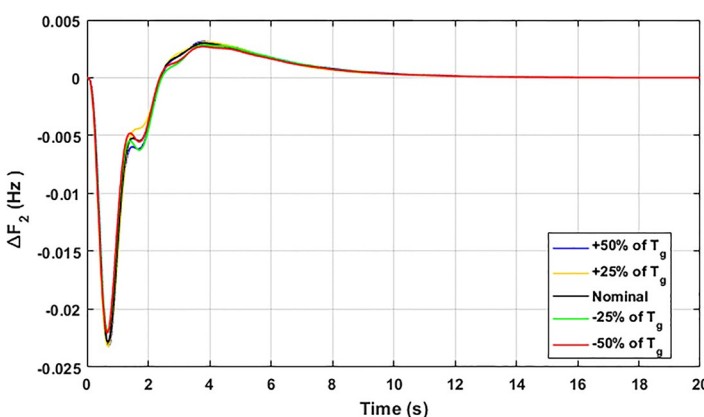

**Fig 37. Results for variation in $T_g$ with $\Delta F_2$.**

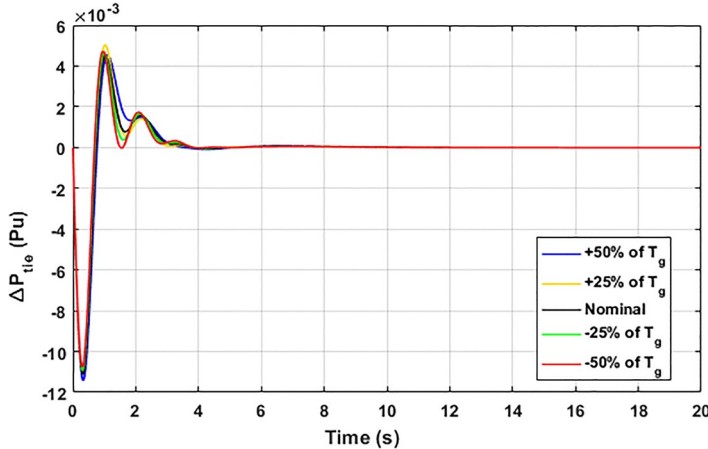

**Fig 38. Results for variation in $T_g$ with $\Delta P_{tie}$.**

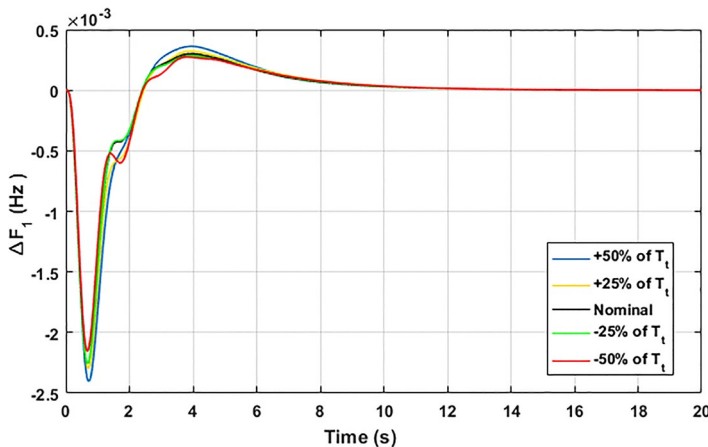

**Fig 39. Results for variation in $T_t$ with $\Delta F_1$.**

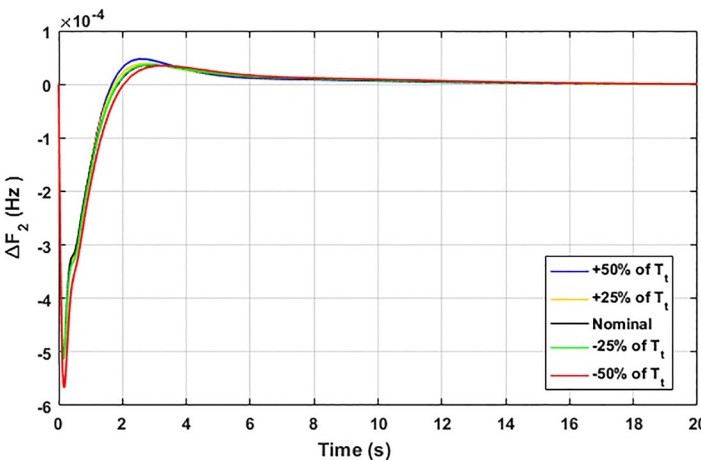

**Fig 40. Results for variation in $T_t$ with $\Delta F_2$.**

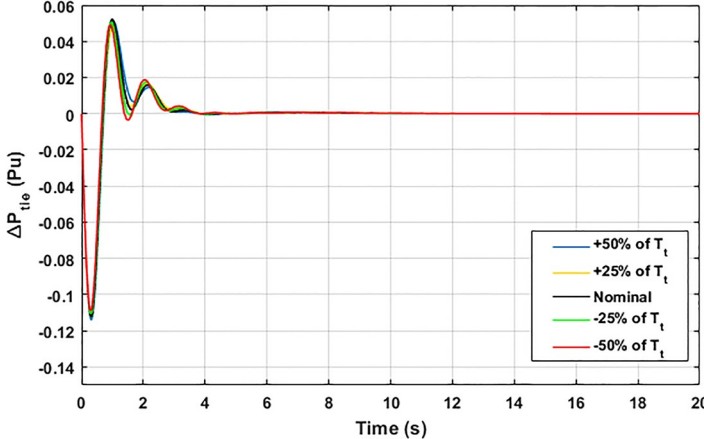

**Fig 41. Results for variation in $T_t$ with $\Delta P_{tie}$.**

parameters are very close to the nominal values and hence, the controller gains need not be re-tuned for a change of system parameters and load conditions within a range of ± 50%.

## 4 Conclusion

In this paper, PID and I-PD controllers have been designed and successfully implemented to handle Automatic Generation Control (AGC) of multi-source with multi-area Interconnected Power System (IPS). Fitness Dependent Optimizer (FDO) algorithm has been used to attain the gains of these controllers. The effectiveness of FDO based controllers i.e. FDO-PID and FDO-I-PD are evaluated on two-area with reheat thermal, gas and hydro power system individually and then collectively with all three generation units present in the system. The transient response performance achieved by the designed controllers with 1% step load perturbation are presented and quantified in detail via simulations and compared with several other controller techniques. The comparison of results evidently shows that FDO-PID and FDO-I-PD controllers provide superior results in respect of Settling time (Ts), overshoot (Osh), and undershoot (Ush) for thermal reheat, gas, hydro and multi-source power generation. It is observed th at FDO-I-PD as compared to DE-PID completely eliminates Osh in load frequency of both areas and in tie-line power, while 83.38%, 80.24%, and 77.54% improvement in Ts in area-1 and area-2 load frequency, and tie-line power are achieved respectively. Similarly, FDO-I-PD also provides an improvement of 83.1%, 77.4%, and 87.8% in Ush as compared DE-PID. The results further show that FDO- I-PD as compared to LUS-PID provides significant improvement of 75.24%, 64.28% and 66.98% in Ts of load frequency for$\Delta F_1$, $\Delta F_2$, and $\Delta P_{tie}$ respectively, while an improvement in Ush of 63.23%, 49.43% and 57.58% are achieved for load frequency of $\Delta F_1$, $\Delta F_2$, and $\Delta P_{tie}$ respectively. The supremacy of FDO based PID and I-PD controllers proposed in this work clearly demonstrate the capability of these controllers to tackle the automatic generation control problem effectively with oscillation-free and quick response. In future studies, the performance of the same power system can be improved by employing robust FOPID controller and powerful meta-heuristic algorithms. The same power system may be extended by incorporating with other renewable energy sources.

## Supporting information

**S1 File.**
(PDF)

## Author Contributions

**Conceptualization:** Amil Daraz, Suheel Abdullah Malik, Ihsan Ul Haq.

**Formal analysis:** Ihsan Ul Haq, Khan Bahadar Khan.

**Investigation:** Ihsan Ul Haq.

**Methodology:** Suheel Abdullah Malik.

**Resources:** Ghulam Fareed Laghari.

**Software:** Amil Daraz.

**Supervision:** Suheel Abdullah Malik.

**Validation:** Amil Daraz.

**Visualization:** Ihsan Ul Haq, Khan Bahadar Khan, Farhan Zafar.

**Writing – original draft:** Amil Daraz.

**Writing – review & editing:** Amil Daraz, Ghulam Fareed Laghari.

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
