## [Decision Letter · Decision Letter 0]

30 Sep 2020

PONE-D-20-28150

Modified PID Controller for Automatic Generation Control of Multi-Source Interconnected Power System Using Fitness Dependent Optimizer Algorithm

PLOS ONE

Dear Dr. Daraz,

Thank you for submitting your manuscript to PLOS ONE. After careful consideration, we feel that it has merit but does not fully meet PLOS ONE’s publication criteria as it currently stands. Therefore, we invite you to submit a revised version of the manuscript that addresses the points raised during the review process.

We look forward to receiving your revised manuscript.

Kind regards,

Wei Yao, Ph.D.

Academic Editor

PLOS ONE

Journal Requirements:

4. We note you have included a table to which you do not refer in the text of your manuscript. Please ensure that you refer to Table 7 in your text; if accepted, production will need this reference to link the reader to the Table.

Reviewers' comments:

Reviewer's Responses to Questions

**Comments to the Author**

1. Is the manuscript technically sound, and do the data support the conclusions?

Reviewer #1: Yes

Reviewer #2: Partly

2. Has the statistical analysis been performed appropriately and rigorously? 

Reviewer #1: Yes

Reviewer #2: No

3. Have the authors made all data underlying the findings in their manuscript fully available?

Reviewer #1: Yes

Reviewer #2: Yes

4. Is the manuscript presented in an intelligible fashion and written in standard English?

Reviewer #1: Yes

Reviewer #2: Yes

5. Review Comments to the Author

Reviewer #1: 1. Authors should clearly clarify the motivation and contribution of this work;

2. Meta-heuristic algorithms are widely and popularly used in PID control gains tuning, authors should address this important domain in Introduction with a thorough discussion, see: (a) Grouped grey wolf optimizer for maximum power point tracking of doubly-fed induction generator based wind turbine, Energy Conversion and Management. 2017. (b) Democratic joint operations algorithm for optimal power extraction of PMSG based wind energy conversion system, Energy Conversion and Management. 2018. and (c) Perturbation observer based fractional-order PID control of photovoltaics inverters for solar energy harvesting via Yin-Yang-Pair optimization, Energy Conversion and Management. 2018.

3. Future studies should be provided in Conclusion;

4. Authors are suggested to consider fractional-order PID which can significantly improve the overall control performance, an additional discussion section should be made on this domain, e.g., Robust fractional-order PID control of supercapacitor energy storage systems for distribution network applications: A perturbation compensation based approach. Journal of Cleaner Production. 2021.

Reviewer #2: In this paper, a modified form of the PID controller known as the I-PD controller is developed for AGC of the two-area multi-source IPS. The FDO algorithm is employed. My comments for the article are as follows.

1. The literature review only describes the work of the existing literature, however, the relationship of the existing literature with this paper is not described. What the developing trend for these literature should also be explained?

2. “The effectiveness of the proposed approach has been assessed on a two-area network with individual source including gas, hydro and reheat thermal unit and then collectively with all three sources.” However, the FDO algorithm method seems to have the same effect on different types of sources. Please describe in detail the differences between the methods applied to different types of sources.

3. It can be observed from Fig 33-41 that the proposed controller provides robustness by changing system parameters within a range of ± 25%. However, it is not indicated whether the robustness can be guaranteed when the parameter range becomes larger, so comparative verification is needed.

4. Please carefully check the format of references, such as "2018; 12(5): 585-97. " in [29]. Please change it to" 2018, 12(5): 585-97."

6. PLOS authors have the option to publish the peer review history of their article (what does this mean?). If published, this will include your full peer review and any attached files.

Reviewer #1: No

Reviewer #2: No

---

## [Author Response · Author response to Decision Letter 0]

23 Oct 2020

Reviewer#1, Concern # 1: Authors should clearly clarify the motivation and contribution of this work.

Author response: Thanks, the motivation and contribution of this research work has been updated in the section of introduction.

1) Author action: We updated the manuscript by adding the motivation and contribution of this work on page number 3/19.

Reviewer#1, Concern # 2: Meta-heuristic algorithms are widely and popularly used in PID control gains tuning, authors should address this important domain in Introduction with a thorough discussion, see: (a) Grouped grey wolf optimizer for maximum power point tracking of doubly-fed induction generator based wind turbine, Energy Conversion and Management. 2017. (b) Democratic joint operations algorithm for optimal power extraction of PMSG based wind energy conversion system, Energy Conversion and Management. 2018. and (c) Perturbation observer based fractional-order PID control of photovoltaics inverters for solar energy harvesting via Yin-Yang-Pair optimization, Energy Conversion and Management. 2018.

Author response: Thanks for detailed guidance. The introduction section has been updated and discuss thoroughly as directed by reviewers.

Author action: We updated the manuscript by adding detailed introduction on page 2 and 3.

Reviewer#1, Concern # 3: Future studies should be provided in Conclusion.

Author response: Thanks, future studies have been included in the section of conclusion.

Author action: We updated the manuscript by adding the future studies on Page number 15/19.

Reviewer#1, Concern # 4: Authors are suggested to consider fractional-order PID which can significantly improve the overall control performance, an additional discussion section should be made on this domain, e.g., Robust fractional-order PID control of supercapacitor energy storage systems for distribution network applications: A perturbation compensation based approach. Journal of Cleaner Production. 2021.

Author response: Thanks for valuable suggestions. FOPID controller will be considered in future studies to evaluate the performance of the same power system as well as more complex system incorporating with non-linearities and three area system with other renewable energy sources. However, in this work we have develop modified form of PID controller known as I-PD controller which perform better as compare to conventional PID controller and also with other techniques reported in references [9] and [30]. Further, a more recent powerful meta-heuristic algorithm known as Fitness Dependent Optimizer (FDO) has been used for the optimization of controller parameters.

Reviewer#2, Concern # 1: The literature review only describes the work of the existing literature, however, the relationship of the existing literature with this paper is not described. What the developing trend for these literature should also be explained?

Author response: Thanks for guidance. The relationship of existing literature with this research work has been updated and provided in the section of introduction of the revised manuscript.

Author action: We updated the manuscript by adding introduction on page number 2 and 3. 

Reviewer#2, Concern # 2: The effectiveness of the proposed approach has been assessed on a two-area network with individual source including gas, hydro and reheat thermal unit and then collectively with all three sources. However, the FDO algorithm method seems to have the same effect on different types of sources. Please describe in detail the differences between the methods applied to different types of sources.

Author response: I agree, with the reviewer comments that the effect of FDO techniques are almost similar with each type of generation which shows that our proposed approach performs well for each type of generation. However, each generation source i.e. reheat thermal, hydro and gas power generation have different transfer function which are given in detail in the subsection of reheat thermal, hydro, gas and multi- source power system. In case of multi-source system, the transfer function of each generation are collectively applied which makes the system higher order in terms of transfer function as compared to individual sources. 

Author action: We updated the manuscript by adding detail in the subsections of reheat thermal, hydro, gas and multi –source on page number 7,8,9,10,11 and 12.

Reviewer#2, Concern # 3: It can be observed from Fig 33-41 that the proposed controller provides robustness by changing system parameters within a range of ± 25%. However, it is not indicated whether the robustness can be guaranteed when the parameter range becomes larger, so comparative verification is needed.

Author response: Thanks, the robustness of the proposed controller is verified by changing the system parameters to ± 50% with a step of ± 25% which are depicted in Fig 33-41 of the manuscript. It can be observed from Fig 33-41 that system response plotted with variation in various parameters are very closed to the nominal values and hence, the controller gains needs not be re-tuned for a change of system parameters within a range of ± 50% which shows the robustness of the proposed controller.

Author action: We updated the manuscript by adding updated figures (Fig 33-41).

Reviewer#2, Concern # 4: Please carefully check the format of references, such as "2018; 12(5): 585-97. " in [29]. Please change it to" 2018, 12(5): 585-97.".

Author response: Thanks for correction and reminding. The references have been thoroughly checked and correct it.

Author action: We updated the manuscript by updating references and correct it according to format.

---

## [Decision Letter · Decision Letter 1]

3 Nov 2020

Modified PID Controller for Automatic Generation Control of Multi-Source Interconnected Power System Using Fitness Dependent Optimizer Algorithm

PONE-D-20-28150R1

Dear Dr. Daraz,

We’re pleased to inform you that your manuscript has been judged scientifically suitable for publication and will be formally accepted for publication once it meets all outstanding technical requirements.

Kind regards,

Wei Yao, Ph.D.

Academic Editor

PLOS ONE

Additional Editor Comments (optional):

Reviewers' comments:

Reviewer's Responses to Questions

**Comments to the Author**

1. If the authors have adequately addressed your comments raised in a previous round of review and you feel that this manuscript is now acceptable for publication, you may indicate that here to bypass the “Comments to the Author” section, enter your conflict of interest statement in the “Confidential to Editor” section, and submit your "Accept" recommendation.

Reviewer #1: All comments have been addressed

Reviewer #2: All comments have been addressed

2. Is the manuscript technically sound, and do the data support the conclusions?

Reviewer #1: Yes

Reviewer #2: Yes

3. Has the statistical analysis been performed appropriately and rigorously? 

Reviewer #1: Yes

Reviewer #2: Yes

4. Have the authors made all data underlying the findings in their manuscript fully available?

Reviewer #1: Yes

Reviewer #2: Yes

5. Is the manuscript presented in an intelligible fashion and written in standard English?

Reviewer #1: Yes

Reviewer #2: Yes

6. Review Comments to the Author

Reviewer #1: (No Response)

Reviewer #2: The revised manuscript has been greatly improved. All my concerns have been addressed and the paper is recommended for publication.

7. PLOS authors have the option to publish the peer review history of their article (what does this mean?). If published, this will include your full peer review and any attached files.

Reviewer #1: No

Reviewer #2: No

---

## [Editor Report · Acceptance letter]

5 Nov 2020

PONE-D-20-28150R1 

Modified PID Controller for Automatic Generation Control of Multi-Source Interconnected Power System Using Fitness Dependent Optimizer Algorithm 

Dear Dr. Daraz:

I'm pleased to inform you that your manuscript has been deemed suitable for publication in PLOS ONE. Congratulations! Your manuscript is now with our production department. 

Kind regards, 

on behalf of

Professor Wei Yao 

Academic Editor

PLOS ONE